# Robust inference of kinase activity using functional networks

Serhan Yılmaz [1✉], Marzieh Ayati[2], Daniela Schlatzer[3], A. Ercüment Çiçek[4,5], Mark R. Chance[3,6] & Mehmet Koyutürk[1,3]

Mass spectrometry enables high-throughput screening of phosphoproteins across a broad range of biological contexts. When complemented by computational algorithms, phospho-proteomic data allows the inference of kinase activity, facilitating the identification of dys-regulated kinases in various diseases including cancer, Alzheimer's disease and Parkinson's disease. To enhance the reliability of kinase activity inference, we present a network-based framework, RoKAI, that integrates various sources of functional information to capture coordinated changes in signaling. Through computational experiments, we show that phos-phorylation of sites in the functional neighborhood of a kinase are significantly predictive of its activity. The incorporation of this knowledge in RoKAI consistently enhances the accuracy of kinase activity inference methods while making them more robust to missing annotations and quantifications. This enables the identification of understudied kinases and will likely lead to the development of novel kinase inhibitors for targeted therapy of many diseases. RoKAI is available as web-based tool at http://rokai.io.

[1] Department of Computer and Data Sciences, Case Western Reserve University, Cleveland, OH, USA. [2] Department of Computer Science, University of Texas Rio Grande Valley, Edinburg, TX, USA. [3] Center for Proteomics and Bioinformatics, Case Western Reserve University, Cleveland, OH, USA. [4] Department of Computer Engineering, Bilkent University, Ankara, Turkey. [5] Department of Computational Biology, Carnegie Mellon University, Pittsburgh, PA, USA. [6] Department of Nutrition, Case Western Reserve University, Cleveland, OH, USA. ✉email: serhan.yilmaz@case.edu

Protein phosphorylation is a ubiquitous mechanism of posttranslational modification observed across cell types and species, and plays a central role in cellular signaling. Phosphorylation is regulated by networks composed of kinases, phosphatases, and their substrates. Characterization of these networks is becoming increasingly important in many biomedical applications, including identification of novel disease-specific drug targets, development of patient-specific therapeutics, and prediction of treatment outcomes[1,2].

In the context of cancer, identification of kinases plays a key role in the pathogenesis of specific cancers and their subtypes, leading to the development of kinase inhibitors for targeted therapy[3–6]. Disruptions in the phosphorylation of various signaling proteins have also been implicated in the pathophysiology of various other diseases, including Alzheimer's disease[7,8], Parkinson's disease[9], obesity and diabetes[10,11], and fatty liver disease[12], among others. As a consequence, there is increased attention to monitoring the phosphorylation levels of phosphoproteins across a wide range of biological contexts and inferring changes in kinase activity under specific conditions.

Mass spectrometry (MS) provides unprecedented opportunities for large-scale identification and quantification of phosphorylation levels[13]. Typically, thousands of sites are identified in a single MS run. Besides enabling the characterization of the changes in the activity of phosphoproteins, MS-based phospho-proteomic data offers insights into kinase activity based on changes in the phosphorylation of known kinase–substrates[14,15]. Observing that phosphorylation levels of the substrates of a kinase offer clues on kinase activity,[14] use a Kolmogorov–Smirnov (K–S) statistic to compare the phosphorylation distributions of substrate sites and all other phosphosites. Building on this idea, kinase–substrate enrichment analysis (KSEA)[15] infers kinase activity based on aggregates of the phosphorylation levels of substrates and assess the statistical significance using Z-test.[16] Develop these ideas further by introducing a heuristic machine learning method, IKAP, which additionally models the dependencies between kinases that phosphorylate the same substrate. Other approaches[17,18] adapt the widely used gene set enrichment analysis (GSEA)[19] for kinase activity inference problem. In parallel to these, a new branch of computational approaches focus on single samples to infer kinase activity[20–23].

Despite the development of algorithms that utilize relatively sophisticated models, KSEA remains one of the most-widely used tools for kinase activity inference[24]. This can be largely attributed to the constraints posed by limited comprehensiveness of available data, prohibiting the utility of such sophisticated models. Available kinase annotations still provide very little coverage (<10%) for phosphosites identified in MS experiments[25]. The coverage of MS-based phospho-proteomics is also limited, and many sites existing in sample may be unidentified due to technical factors[26]. Computationally predicted kinase–substrate associations[27,28] are successfully utilized to expand the scope of kinase activity inference[29]. However, the coverage of computationally predicted associations is also limited[30] and most algorithms can only make predictions for well-studied kinases[31].

With a view to expanding the scope of kinase activity inference, we develop a framework that comprehensively utilizes available functional information on kinases and their substrates. We hypothesize that biologically significant changes in signaling manifest as hyper-phosphorylation or dephosphorylation of multiple functionally related sites. Therefore, having consistently hyper-phosphorylated (or dephosphorylated) sites in the functional neighborhood of a phosphosite can provide further evidence about the changes in the phosphorylation of that site. Our framework, robust kinase activity inference (RoKAI), uses a heterogeneous network model to integrate relevant sources of functional information, including: (i) kinase–substrate associations from PhositePlus[32], (ii) coevolution and structural distance evidence between phosphosites from PTMcode[33], and (iii) protein–protein interactions (PPI) from STRING[34] for interactions between kinases. On this heterogeneous network, we propagate the quantifications of phosphosites to compute representative phosphorylation levels capturing coordinated changes in signaling. To predict changes in kinase activity, we use these resulting representative phosphorylation levels in combination with existing kinase activity inference methods.

In order to increase the coverage of network propagation, we develop an electric circuit-based model[35,36] that is specifically designed to incorporate missing sites not identified by MS. While RoKAI does not impute phosphorylation levels for unidentified sites (i.e., it is not intended to fill in missing data), it uses these sites to bridge the functional connectivity among identified sites. Similar electric circuit-based models have been employed in the analysis of expression quantitative trait loci to identify causal genes and dysregulated pathways[37,38]. However, one important distinction is that the electric circuit model in RoKAI does not aim to uncover intermediate nodes between select target nodes, rather, it propagates all available quantifications over the network in order to reduce the noise by capturing consistent changes in the functional neighborhood of every node.

A recent study by[39] benchmarks substrate-based inference approaches using a comprehensive atlas of human kinase regulation[18], encompassing more than fifty perturbations. Using this dataset, we systematically benchmark the improvement provided by RoKAI on the performance of a variety of kinase activity inference methods. In our computational experiments, we observe that the benchmark data is substantially biased in favor of "rich kinases" with many known substrates. Our results show that methods that appear to provide superior performance (e.g., methods that utilize statistical significance) accomplish this by increasing bias toward such rich kinases (since statistical power goes up with increasing number of observations). Motivated by this observation, we systematically evaluate the robustness of kinase activity inference methods using Monte Carlo simulations with varying levels of missingness. The results of this analysis shows that methods biased toward rich kinases are more vulnerable to incompleteness of available kinase–substrate annotations.

Next, we characterize the contribution of each source of functional information on enhancing kinase activity inference. Our results show that incorporation of "shared-kinase associations" (i.e., transferring information between sites that are targeted by the same kinase) significantly improves kinase activity inference. We observe that, other sources of functional information considered (PPI, coevolution, and structure distance evidence) also provide statistically significant information for kinase activity inference. However, their contribution is smaller in comparison due to either (i) limited coverage or (ii) redundancy with existing kinase–substrate annotations. Finally, we systematically investigate the performance of RoKAI in improving the performance of kinase activity methods. Results of these computational experiments show that RoKAI consistently improves the accuracy, stability, and robustness of all kinase activity inference methods that are benchmarked.

Overall, our results clearly demonstrate the utility of functional information in expanding the scope of kinase activity inference and establish RoKAI as a useful tool in pursuit of reliable kinase activity inference. RoKAI is available as a web tool (http://rokai.io), as well as an open source MATLAB package (http://compbio.case.edu/omics/software/rokai).

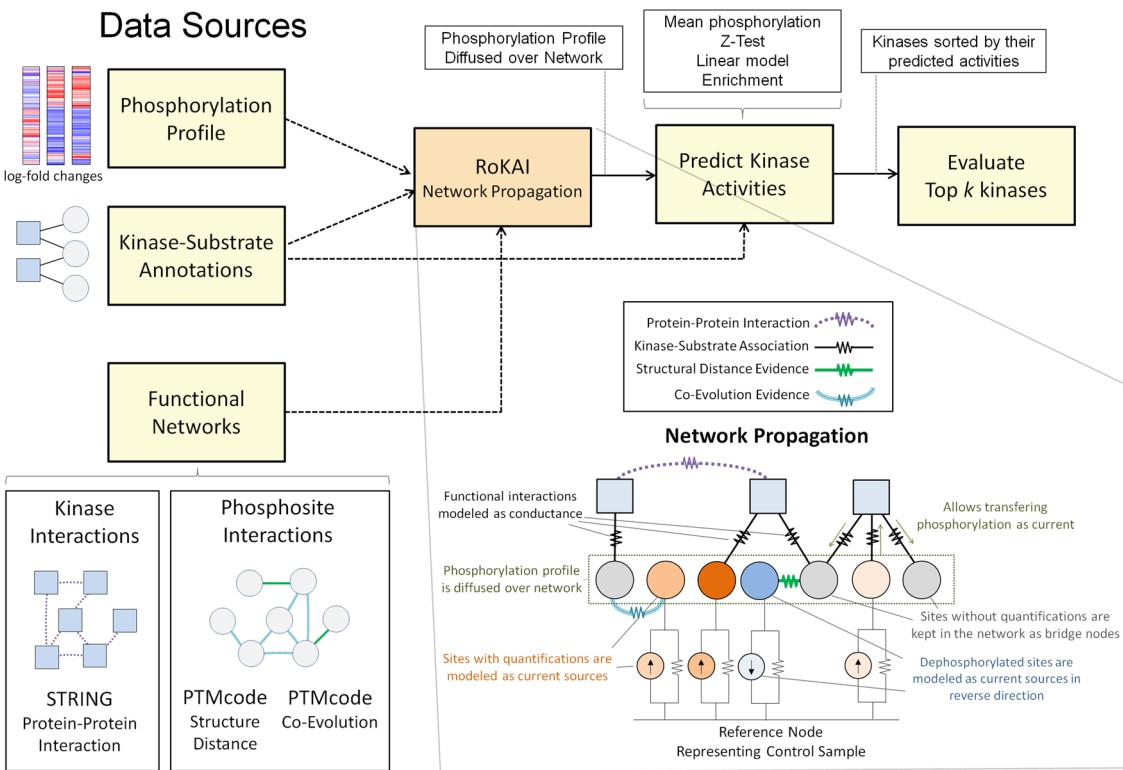

**Fig. 1 The workflow and the key idea of RoKAI.** Traditional algorithms for kinase activity inference use condition-specific phosphorylation data and available kinase–substrate associations to identify kinases with differential activity in each condition. RoKAI integrates functional networks of kinases and phosphorylation sites to generate robust phosphorylation profiles. The network propagation algorithm implemented by RoKAI ensures that unidentified sites that lack quantification levels in a condition can still be used as bridges to propagate phosphorylation data through functional paths.

## Results

**Robust inference of kinase activity with RoKAI.** With a view to rendering kinase activity inference robust to missing data and annotations, we develop RoKAI, a network-based algorithm that utilizes available functional associations to compute refined phosphorylation profiles. We hypothesize that biologically significant changes in signaling manifest as hyper-phosphorylation or dephosphorylation of multiple functionally related sites. Therefore, having consistently hyper-phosphorylated (or dephosphorylated) sites in the functional neighborhood of a phosphosite can provide further evidence about the changes in the phosphorylation of that site. Conversely, inconsistency in the change in the phosphorylation levels of sites in a functional neighborhood can serve as negative evidence that can be used to reduce noise.

Based on this hypothesis, we develop a heterogeneous network model (with kinases and phosphosites as nodes) to propagate the phosphorylation of sites across functional neighborhoods. In this model, each edge has a conductance allowing some portion of the phosphorylation to be carried to the connecting nodes (illustrated in Fig. 1). Therefore, the propagated phosphorylation level of a site represents an aggregate of the phosphorylation of the site and the sites that are (directly or indirectly) functionally associated with it. Consequently, the propagated phosphorylation profiles are expected to capture coordinated changes in signaling, which are potentially less noisy and more robust.

It is important to note that, we do not use network propagation to directly infer kinase activity. Rather, we use it to generate refined phosphorylation profiles that are subsequently used as input to a kinase activity inference method. Thus, the framework of RoKAI can be used together with any existing or future inference methods.

**Experimental setup.** In this section, we describe our benchmarking setup for assessing the performance and robustness of kinase activity inference methods. First, we demonstrate the bias in the gold standard benchmarking data and show how this bias can lead to misleading conclusions on the performance of existing methods. Next, we introduce a robustness analysis in order to (i) overcome the effect of bias on performance estimations, and (ii) to assess the reliability of these algorithms in the presence of missing data. To characterize the value added by RoKAI, we start by assessing the utility of different sources of functional information in inferring kinase activity. Next, by focusing on a baseline kinase activity inference method (mean substrate phosphorylation), we systematically assess the incorporation of various networks with RoKAI in enhancing the accuracy and robustness of the inference. We then assess the generalizability of these results to a broad range of kinase activity inference methods. Afterward, we investigate whether RoKAI's ability to incorporate missing sites in its functional network contributes to the improvement of kinase activity inference. Finally, we explore the effect of including predicted kinase–substrate associations within the RoKAI's framework.

### Benchmarking setup

*Benchmarking data.* Ref. [18] compiled phospho-proteomics data from a comprehensive range of perturbation studies and used these data to comprehensively benchmark the performance of kinase activity inference methods[39]. This benchmark data brings together 24 studies spanning 91 perturbations that are annotated with at least one upregulated or downregulated kinase. In each of the studies, the phosphorylation levels of phosphosites are quantified using MS. After applying quality control steps (as described in the "Methods"), we analyze a subset of this dataset

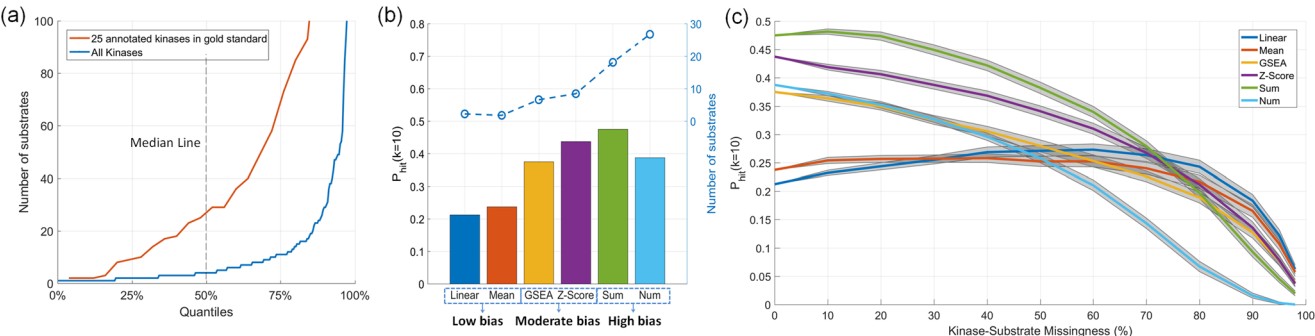

**Fig. 2 Existing benchmark data for kinase activity inference is biased toward kinases with high number of substrates and can be misleading in assessing the performance of inference methods. a** Inverse cumulative distribution of the number of substrates for the 25 kinases that are annotated with a perturbation in gold standard benchmarking data compared to all kinases. The x-axis indicates the quantiles. For example, the value on the y-axis that corresponds to $x = 50\%$ indicates the median number of substrates. **b** Performance and bias of baseline kinase activity inference methods. The bars show the probability of identifying an annotated "true" kinase in top ten predicted kinases ($P_{hit}$10). The dashed line indicates the average number of substrates of the top ten predicted kinases for the corresponding method. The high-bias methods (Sum: total substrate phosphorylation, and Num: number of substrates) are not used in the literature, but are shown here to illustrate the effect of bias on performance assessment. **c** The robustness analysis of the methods for missingness in kinase–substrates links. The x-axis shows the percentage of (randomly selected) kinase–substrates links of 25 gold standard kinases hidden from the kinase activity inference methods. The gray areas indicate the 95% confidence intervals for the mean performance across 100 runs.

encompassing 80 perturbations and 53,636 phosphosites identified in at least one of these 80 perturbations. Overall, for these 80 perturbations, there are 128 kinase-perturbation annotations (which is considered gold standard) for 25 different kinases (listed in Supplementary Data 1). In our computational experiments, we use this dataset to assess the robustness of existing kinase activity inference methods and validate our algorithms.

*Kinase–substrate annotations.* We obtain existing kinase–substrate associations from PhosphositePlus[32]. PhosphositePlus contains a total of 10,476 kinase–substrate links for 371 distinct kinases and 7480 sites. Among these annotated sites, 2397 have quantifications in the perturbation data. These sites have a total of 3877 kinase–substrate links with 261 kinases.

*Benchmarking metric.* The main purpose of kinase activity inference is to prioritize kinases for additional consideration and ideally for experimental validation. However, in practice, it is typically very costly to experimentally validate more than a few kinases[40] and it is infeasible to manually inspect more than a couple dozen. Whereas, benchmarking approaches that are employed in the literature like area under receiver operating characteristics curve and precision at recall 0.5 consider high number of predictions ($k$), including kinases that are less significant/active. We find such measures problematic because, even though they include the performance for a high number of predicted kinases in their calculation, it would not be practical for a potential user to inspect or use that many predictions. To take this consideration into account, we use a metric, "top-$k$-hit", that focuses on the top $k$ kinase predictions for small values of $k$. Since the gold standard dataset is incomplete, this metric essentially serves as a minimum bound on the expected probability of discovering an upregulated or downregulated kinase if top $k$ kinases predicted by the inference method were to be experimentally validated. In our experiments, we use $k = 10$ (unless otherwise specified) since it represents a reasonable number of kinases to be put to additional scrutiny before experimental validation.

*Existing inference methods.* Kinase activity inference methods differ from each other in terms how they integrate the phosphorylation levels of the substrates of a kinase to estimate its activity. These methods range from simple aggregates and

enrichment analyses to more sophisticated methods that take into account the interplay between different kinases. We benchmark the following commonly used inference methods:

- Mean (baseline method): one of the simpliest kinase activity inference methods employed by KSEA[15]. This method represents the activity of a kinase as the mean phosphorylation of its substrates.
- Z-score: to assess the statistical significance of inferred activities, KSEA uses z-scores, normalizing the total log-fold change of substrates with the standard deviation of the log-fold changes of all sites in the dataset.
- Linear model: the linear model, considered by ref. [39], aims to take into account of the dependencies between kinases that phosphorylate the same site. In this model, the phosphorylation of a site is modeled as summation of the activities of kinases that phosphorylate the site. A similar (but more complex) approach is also utilized by IKAP[16].
- GSEA: refs. [17] and [18] adopt GSEA, a widely used method in systems biology[19], to infer kinase activity by assessing whether the target sites of a kinase exhibit are enriched in terms of their phosphorylation fold change compared to other phosphosites.

**Bias and robustness of existing inference methods.** Previous benchmarking by ref. [39] suggests that methods that rely on statistical significance (Z-Score and GSEA) are superior to their alternatives. However, as shown in Fig. 2a, we observe that there is substantial bias in the benchmarking data: "rich" kinases (i.e., kinases with many known substrates) are significantly over-represented among the 25 annotated kinases that have at least one perturbation (median number of substrates: 29 for annotated and 4 for not-annotated kinases, K–S test $p$ value $< 3.5e-7$ for the comparison of annotated kinases with others in terms of their distribution of number of substrates).

Since methods that rely on statistic significance have a positive bias for kinases with many substrates (statistical power is improved with number of observations), we hypothesize that this is the reason behind their observed superior performance. To test this hypothesis, we benchmark two additional inference methods that are artificially biased for kinases with many

substrates: (i) Sum: sum of phosphorylation (log-fold changes) of substrates, and (ii) Num: number of substrates, used directly as the predicted activity of a kinase (clearly, this method does not use the phosphorylation levels of sites, thus, it always generates the same ranking of kinases regardless of the phosphorylation data). As shown in Fig. 2b, methods that are artificially biased for rich kinases appear to have better predictivity over the alternatives.

In order to overcome the effect of this bias on evaluation, we perform a robustness analysis where we hide a percentage of the known substrates of the 25 annotated kinases from the inference methods. The results of this analysis are shown in Fig. 2c. As seen in the figure, even though methods biased for rich kinases appear to have higher predictivity when all of the available kinase–substrate annotations are used, they are not robust to increasing rate of missingness in kinase–substrate annotations. The performance of artificially biased methods fall below that of the low-biased methods (e.g., mean and linear model) at ~50% missingness. At ~80% missingness, the effect of the bias on evaluation is mitigated, i.e., the difference between number of substrates of 25 annotated kinases and the remaining kinases is not at a statistically detectable level anymore. Thus, the performance of biased (e.g., statistical significance based) methods fall below the low-bias methods at ~80% missingness. These observations make the reliability of biased methods highly questionable since the available kinase–substrate annotations are largely incomplete.

**Utility of functional networks for inferring kinase activity**. To improve the predictions of kinase activity inference methods in a robust manner, our approach is to utilize available functional or structural information. We hypothesize that phosphorylation of sites that are related to the kinase–substrates (whether functionally or structurally) would be predictive of kinase activity. Specifically, we investigate the predictive ability of following functional networks:

- Known kinase–substrates (baseline network): this network comprises of the kinase–substrate associations obtained from PhosphoSitePlus. This is the (only) network that is utilized by all kinase activity inference methods and serves as our baseline.
- Shared-kinase interactions: here, we consider two phospho-sites to be neighbors if both are phosphorylated by the same kinase. We hypothesize that phosphorylation of neighbor sites of kinase–substrates would be predictive of kinase activity. Note that in RoKAI's heterogeneous functional network, there are no additional edges that represent shared-kinase interactions. Instead, RoKAI's network propagation algorithm propagates phosphorylation levels across shared-kinase sites through paths composed of kinase–substrate associations.
- STRING PPI: we hypothesize that the phosphorylation levels of the substrates of two interacting kinases will be predictive of each other's activity.
- PTMcode structural distance evidence: we hypothesize that the phosphorylation of sites that are structurally similar to a kinase's substrates will be predictive of that kinase's activity.
- PTMcode coevolution evidence: we hypothesize that phosphorylation of sites that show similar evolutionary trajectories to a kinase's substrates will be predictive of that kinase's activity.

For each of the functional or structural networks described above, we compute a network activity prediction score for each kinase based on the mean phosphorylation of sites that are considered of interest for the corresponding network (illustrated

in Fig. 3). Note that, except for the baseline network (known kinase–substrates), we do not use the phosphorylation levels of the kinase's own substrates to compute the scores for each network.

To characterize the contribution of each source of functional information on enhancing kinase activity inference, we consider the following metrics:

- Predictivity: to assess the utility of functional networks in predicting the "true" perturbed kinases in gold standard dataset, we use K–S test[41] comparing the distribution of network scores for true kinases with the distribution of all other kinases. For each functional network, we consider the K–S statistic as the predictivity score of the corresponding network.
- Coverage: the network scores contain missing values for kinases without any edges in the corresponding functional networks. Thus, while assessing predictivity (as explained above), we utilize only the kinases with a valid network score. To take missing data into account, we compute a coverage score which is equal to the percentage of kinases with a valid network score with respect to that functional network.
- Complementarity: we aim to utilize the functional networks as an information source that complements available kinase–substrate associations. If there is statistical dependency between functional network scores and the activity inferred by the kinase's own sites, the information provided by the network would be redundant. We use complementarity score as one minus absolute linear (Pearson) correlation between the score of each network scores and kinase activity inferred based on the kinase's own substrates. Since the kinase–substrate association network serves as our baseline, we consider it to have 100% complementarity.
- Overall effect: to quantify the overall contribution of the functional networks for improving the predictions of kinase activity, we combine the predicity, coverage and complementarity scores and obtain an overall effect score:

$$\text{Overall effect} = \text{predictivity} \times \text{coverage} \times \text{complementarity} \quad (1)$$

The results of this analysis are shown in Fig. 3. As can be seen, all considered functional information sources exhibit statistically significant predictivity of the kinase-perturbations according to two-sample K–S test: known kinase–substrates (K–S statistic = 0.21, $p$ value ≤ 1.3e−4), shared-kinase interactions (K–S statistic = 0.21, $p$ value ≤ 7.3e−5), PPI (K–S statistic = 0.18, $p$ value ≤ 8.7e−4), structure distance evidence (K–S statistic = 0.29, $p$ value ≤ 0.03), and coevolution evidence (K–S statistic = 0.26, $p$ value ≤ 5.5e−5). We observe that the incorporation of "shared-kinase associations" in addition to the known kinase–substrates has the most overall contribution to the inference of kinase activities (Fig. 3a, b), followed by kinase–kinase interactions (Fig. 3c). Even though coevolution and structural distance networks exhibit strong predictivity, their overall contribution is relatively low due to their limited coverage and redundancy with existing kinase–substrate annotations (Fig. 3d, e).

**Benchmarking RoKAI-enhanced inference methods**. Motivated by the utility of the functional networks for predicting kinase activity, we gradually explore a set of heterogeneous networks with RoKAI by adding sources of functional information primarily based on their overall effect observed in the previous section:

- Kinase–substrate (KS) network: the network used by RoKAI consists only of the known kinase–substrate interactions. Use of this network allows RoKAI to utilize sites with shared-kinase interactions (illustrated in Fig. 3b), i.e., sites that are

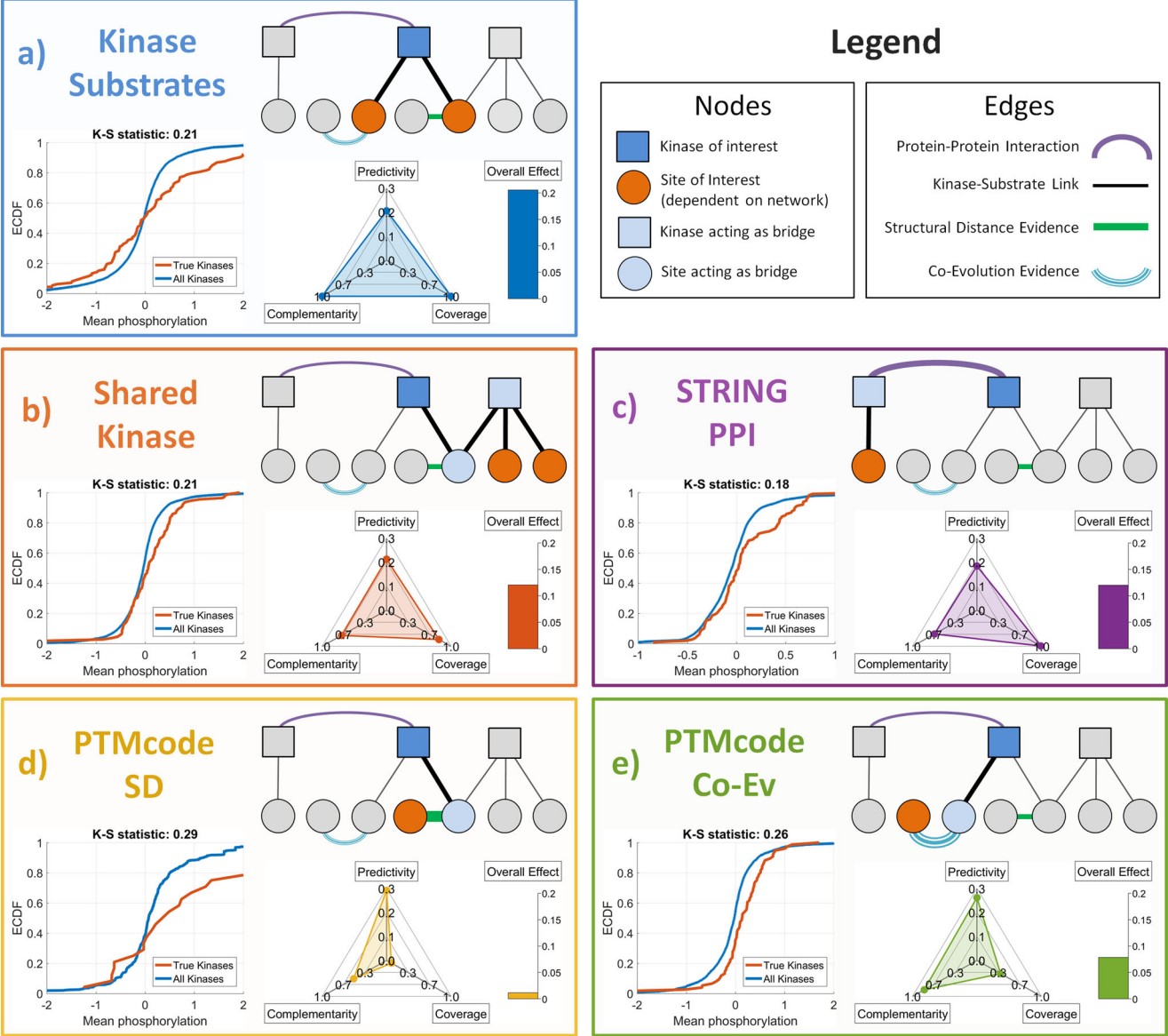

**Fig. 3 Utility of available functional or structural information in providing information on kinase activity.** Each panel (labeled **a**–**e**) represents a different information source. The first panel (kinase–substrates) represents the information source that is utilized by all existing kinase activity inference methods, whereas, the other four panels represent the information sources introduced here. In each panel, the relationship between a kinase (blue square) and the site(s) (red circles) that provide(s) information on the activity of the kinase is illustrated. The bottom-left plot compares the empirical cumulative distribution (ECDF) of the phosphorylation levels of the "information-providing" sites for "true" perturbed kinases in the benchmark data against all kinases. The bottom-right plot shows the predictivity (accuracy in predicting kinase activity), complementarity (information provided in addition to the substrates of the kinase), and coverage (fraction of kinases that are affected) of the information source. The bars represent the overall effect of the information source calculated as the product of the scores shown on the other axes.

targeted by the same kinase contribute to their refined phosphorylation profiles.

- KS + PPI network: in addition to KS, this network includes weighted PPI between kinases. This allows propagation of phosphorylation levels between substrates of interacting kinases (illustrated in Fig. 3c).
- KS + PPI + SD network: in addition KS + PPI, this network includes interactions between phosphosites with structural distance (SD) evidence obtained from PTMcode. This allows the utilization of sites that are structurally proximate to the substrates of a kinase (illustrated in Fig. 3d).
- KS + PPI + SD + CoEv (combined) network: in addition KS + PPI + SD, this network includes interactions between

phosphosites with coevolution evidence obtained from PTMcode. This allows the utilization of sites that are evolutionarily similar to the substrates of a kinase (illustrated in Fig. 3e).

To assess the performance of RoKAI with these networks, we use the benchmarking data from the atlas of kinase regulation. As previously discussed, this dataset is heavily biased toward kinases with many known substrates. To overcome the effect of this bias on evaluation, we perform robustness analyses where we hide a portion of known kinase–substrate interactions of the 25 kinases that have perturbations. For predicting kinase activity, we use the mean substrate phosphorylation (baseline inference method) and

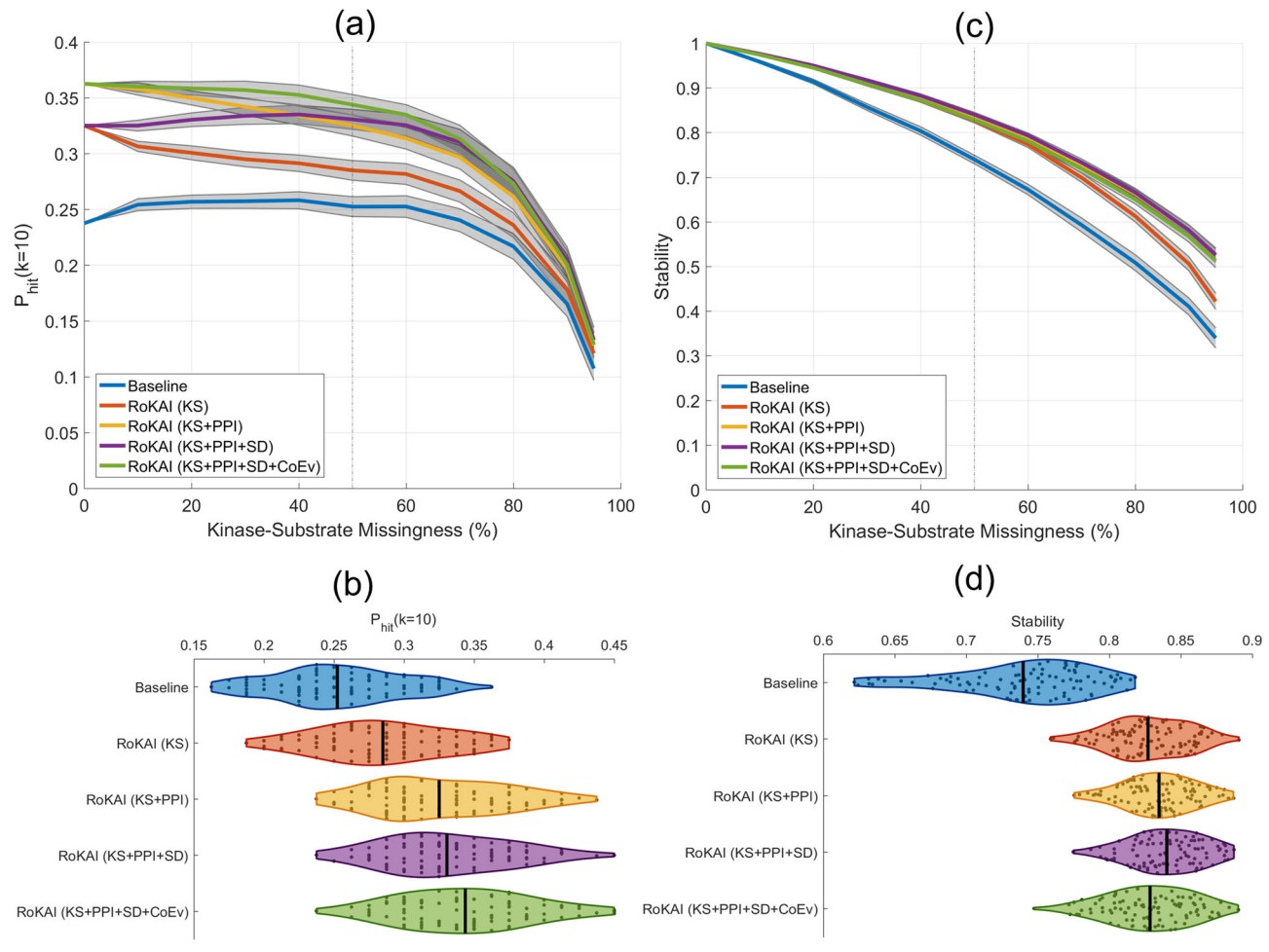

**Fig. 4 Comparison of the accuracy and stability of mean substrate phoshorylation and its RoKAI-enhanced versions using various functional or structural networks. a** The hit-10 performance (the probability of ranking a true perturbed kinase in the top ten), as a function of missingness (the fraction of kinase–substrate associations that are hidden). The shaded areas indicate the 95% confidence intervals for the mean performance across 100 randomized runs. **b** The distribution of hit-10 probabilities for 100 runs at 50% missingness. **c** Stability of the inferred activities (measured by the average squared correlation between inferred activities when different portions of kinase–substrate associations are hidden from the inference methods) as a function of missingness. The shaded areas indicate 95% confidence intervals for the mean stability across 100 randomized runs. **d** The distribution of stability for 100 runs at 50% missingness.

compare the performance of original predictions and RoKAI-enhanced predictions. As shown in Fig. 4a, b, RoKAI consistently and significantly ($p < 0.05$) improves the predictions in a robust manner for varying levels of missing data.

The functional networks that contribute most to the improvements in prediction performance of RoKAI are respectively: KS network (modeling shared-kinase interactions) followed by PPI (for including kinase–kinase interactions) followed by coevolution evidence. Compared to these, including structural distance evidence in the network has a minor effect on prediction performance. This is in line with the overall effect size estimations (shown in Fig. 3). Since structural distance network has relatively small number of such edges, it provides low coverage and a minor effect size even though the existing edges are estimated to be more predictive of kinase activity compared to other networks.

To further evaluate the robustness of the predictions, we assess the stability, i.e., the expected degree of aggreement between the predicted kinase activity profiles when different kinase–substrates are used (e.g., because some sites are not identified by a MS run)

to infer the activity of a kinase. We measure the stability by computing average squared correlation between different runs of robustness analysis (where a different portion of kinase–substrate links are used for inferring kinase activity in each run). As shown in Fig. 4c, d, predictions made by RoKAI-enhanced phosphorylation profiles are significantly ($p < 0.05$) more stable in addition to being more predictive.

**Improvement of RoKAI over a broad range of methods**. Since RoKAI provides refined phosphorylation profiles (propagated by functional networks), it can be used in conjunction with any existing (or future) kinase activity inference algorithms. Here, we benchmark the performance of RoKAI when used together with existing inference methods. For each of these methods, we use the refined phosphorylation profile (obtained by RoKAI) to obtain the RoKAI-enhanced kinase activity predictions. To assess the prediction performance while addressing the bias for rich kinases, we perform robustness analysis at 50% kinase–substrate missingness and measure the top-$k$-hit

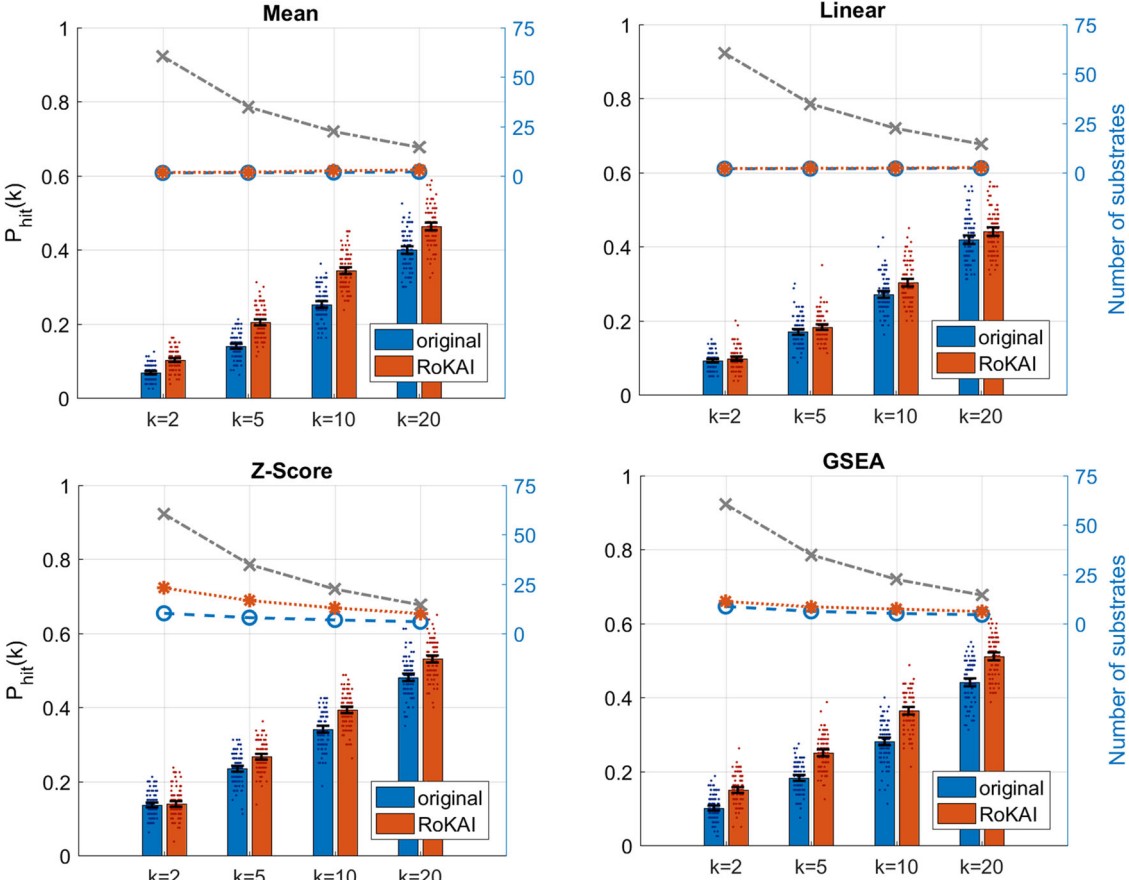

**Fig. 5 Contribution of RoKAI (combined network) in improving the performance of different kinase activity inference methods for predicting the true (annotated) kinase in the top _k_ kinase predictions for various _k_.** The bars show the mean probability of predicting a true kinase among the top _k_ kinases at 50% kinase–substrate missingness. The blue bars indicate the prediction performance using the original (unmodified) phosphorylation profiles and red bars indicate the performance of using RoKAI-enhanced profiles for inferring kinase activity. The colored dashed lines indicate the average number of substrates of the top _k_ kinases predicted by the corresponding inference method (the gray dashed line shows the maximum possible). The black error bars indicate the 95% confidence intervals for the mean performance across 100 randomized runs. The colored points around each bar indicate the performance on different runs.

performance for $k = 2$, 5, 10, and 20. As shown in Fig. 5, RoKAI consistently improves the predictions of all inference methods tested.

**Effect of incorporating unidentified sites in RoKAI.** An important feature of RoKAI's network propagation algorithm is its ability to accommodate unidentified sites (i.e., sites that do not have quantified phosphorylation levels in the data) in the functional network. While RoKAI does not impute phosphorylation levels for unidentified sites (i.e., it is not intended to fill in missing data), it uses these sites to bridge the functional connectivity among identified sites. To assess the value added by this feature, we compare two versions of RoKAI: one that removes unidentified sites from the network (type I) and one that utilizes unidentified sites as bridges (type II). The results of this analysis are shown in Fig. 6. The kinase activity inference activity method we use in these experiments is mean phosphorylation level. As seen in the figure, retention of unidentified sites in the network consistently improves the accuracy of kinase activity inference although the magnitude of this improvement is rather modest (in comparison to the overall improvement of RoKAI to the baseline). We observe a similar improvement for all other kinase activity inference methods that are considered.

**Effect of incorporating predicted kinase–substrate associations.** Next, we investigate the utility of using predicted kinase–substrate associations within the RoKAI's framework. For this purpose, we use NetworKIN[28], which lists its predictions separately as motif-based (NetPhorest), interaction-based (STRING), or combined (using both motif and interaction informations). To incorporate these predictions in RoKAI's framework, we consider two strategies:

1. Include the predicted kinase–substrate interactions (in addition to known substrates in PhosphositePlus) during the kinase activity inference but do not alter the RoKAI's functional network.
2. Include the predicted interactions in both RoKAI's functional network and during the kinase activity inference (this strategy is annotated RoKAI+).

For this analysis, we use the baseline method (mean phosphorylation) for the inference. To make the results comparable with our previous analysis (using only the known substrates in PhosphositePlus), we limit the analysis to the kinases with at least one known substrate identified in the perturbation experiments (this way, we keep the kinase set same as before). The results of this analysis are shown in Fig. 7. Here, the _x_-axis shows the number of predicted interactions included in

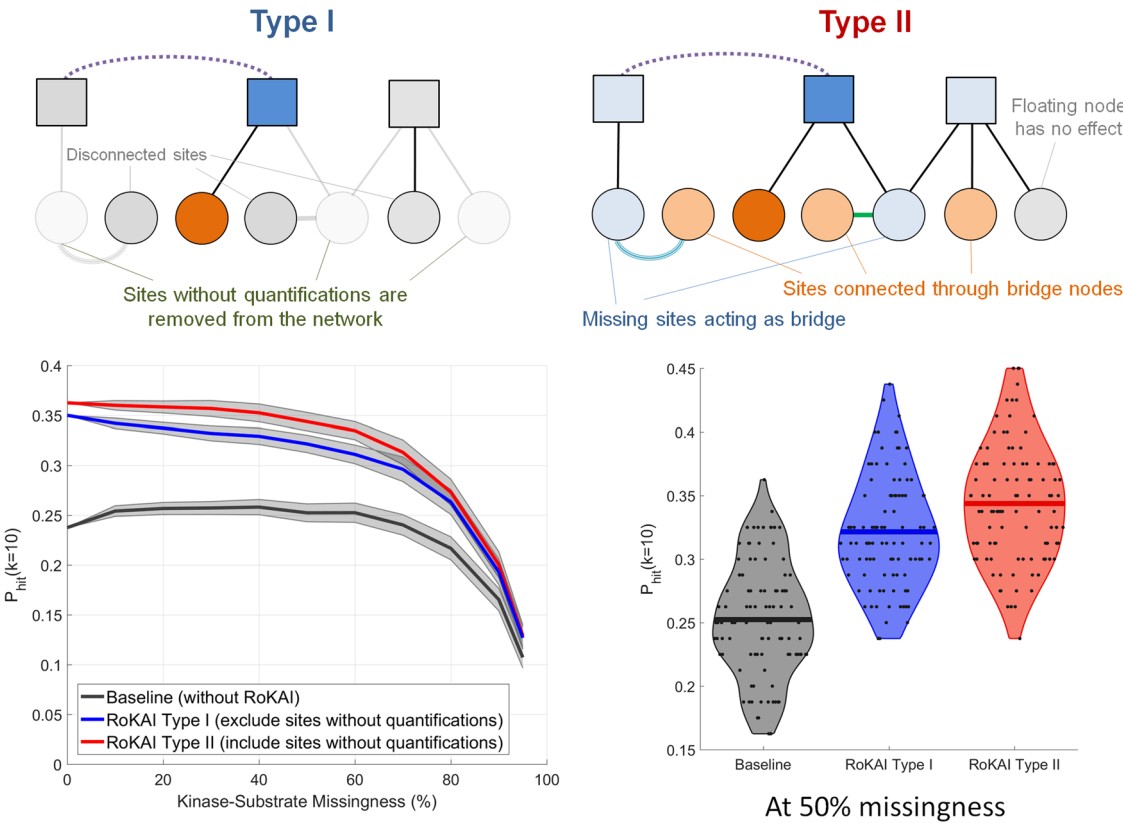

**Fig. 6 RoKAI improves kinase activity inference by enabling utilization of the unidentified sites (without quantifications) for predicting the activity of kinases.** In type I (illustrated in top left), the network consists only of sites with quantifications. Whereas, in type II (illustrated in top right), the network includes sites without quantifications to utilize them as bridge nodes. (Bottom left) robustness analysis with respect to missingness of kinase–substrate links. The shaded area shows the 95% confidence interval for the mean performance on 100 randomized runs where different kinase–substrate links are removed. (Bottom right) The performance of RoKAI type I and type II at 50% missingness. Each point indicate the performance on a different run. The lines indicate the mean performance across 100 runs.

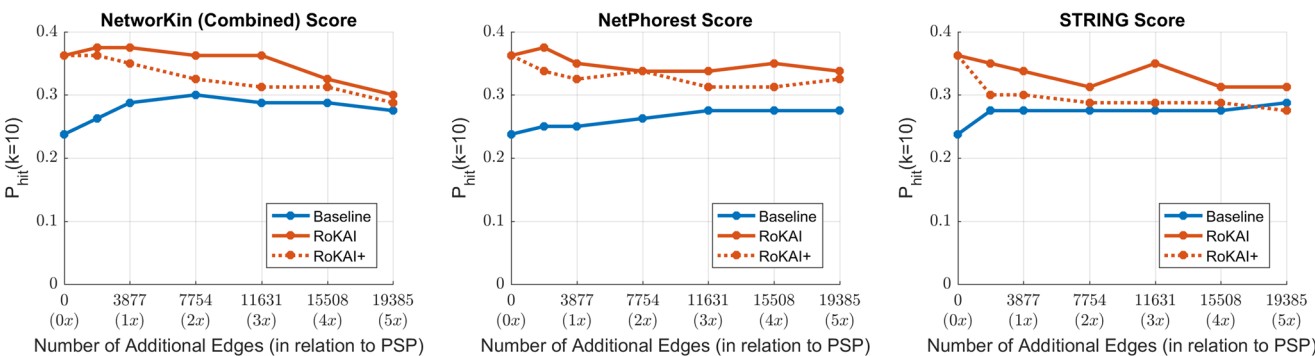

**Fig. 7 The effect of including kinase–substrate links predicted by NetworKin on kinase activity inference.** Each panel shows the results for a difference scoring (used for kinase–substrate edges). In each panel, the x-axis shows the number of edges used by the inference methods in addition to the kinase–substrate annotations from PhosphositePlus (PSP). The colored blue and orange lines indicate the performance of baseline method (mean substrate phosphorylation) and its RoKAI-enhanced version, respectively. The dashed-orange line (RoKAI+) indicate the performance when the functional network of RoKAI additionally includes the predicted kinase–substrate edges by NetworKin.

the inference (i.e., as we go right on the x-axis, we apply a more relaxed threshold on the prediction score). Thus, the leftmost point (0 at x-axis) corresponds to the case where only confirmed interactions (PhosphositePlus) are used.

As expected, the inclusion of predicted interactions in kinase activity inference improves the performance for the baseline algorithm and the performance of RoKAI-enhanced inference

stays above the baseline. However, we observe that the use of predicted interactions together with RoKAI does not improve the inference further (while there is some increase in performance with the inclusion of high-confidence predictions, the inclusion of lower-confidence predictions degrades the performance). In addition, the inclusion of predicted interactions within the RoKAI's functional network always results in less accurate

inference. Taken together, these observations suggest that, since RoKAI already includes functional and structural information to compute propagated phosphorylation levels, the inclusion of predicted interactions that use similar information does not further enhance the accuracy of the inference.

## Discussion

By comprehensively utilizing available data on the functional relationships among kinases, phosphoproteins, and phosphorylation sites, RoKAI improves the robustness of kinase activity inference to the missing annotations and quantifications. Its implementation is available as open source in Matlab, as well as a web tool (http://rokai.io) for easy accessibility. We expect that this will facilitate the identification of understudied kinases with only a few annotations and lead to the development of novel kinase inhibitors for targeted therapy of many diseases, such as cancer, Alzheimer's disease, and Parkinson's disease. As additional functional information on cellular signaling becomes available, the inclusion of these information in functional networks utilized by RoKAI will likely further enhance the accuracy and robustness of kinase activity inference.

The introduced benchmarking setup provides the opportunity to explore and compare the predictions of a variety of inference algorithms in terms of their robustness to missing annotations. It also allows the estimation of how utilization of different functional networks would influence the inference process. These features can help enable researchers to understand the trade-offs between different kinase activity inference algorithms in terms of their robustness, accuracy, and biases. As a potential resource, we provide the materials (code and data) to reproduce our analysis results in figshare (https://doi.org/10.6084/m9.figshare.12644864) that the users can adapt to test different inference methods and/or networks. Using such a framework, we believe the users can make more informed decisions for follow-up studies.

A noteworthy complication in perturbation studies that concern kinase activity inference is the effect of off-target kinases. While recent studies systematically identify off-target kinases in perturbation studies[42,43], the extension of kinase activity inference algorithms and tools like RoKAI to distinguish off-target effects remains an open problem that can advance many important applications like the drug development.

An important consideration in kinase activity inference is the dependencies between phosphorylation levels of sites. Some inference methods take into account the dependency between sites that are targeted by the same kinase[16,39]. On the other hand, recent studies utilize protein expression to take into account the dependency between sites on the same protein by normalizing phosphorylation levels of the sites, but results on the effectiveness of this approach are not conclusive[44,45]. Whereas, RoKAI implicitly considers the dependencies between sites using a functional network model. We recognize the explicit modeling of the dependencies as an important open problem that can further enhance the performance and reliability of kinase activity inference.

A key motivation in developing RoKAI was to utilize the missing sites without quantifications by keeping them as bridges in the network (thus, increasing the overall coverage of the network). In our experiments, we indeed observe a consistent improvement for incorporating missing sites (as compared to disregarding them completely). However, contrary to our expectation, the magnitude of this improvement is rather modest. We hypothesize that this may be because of (i) biological redundancy, i.e., sites that are reached by missing, bridge nodes may already be covered by other paths consisting of identified

nodes, and (ii) our incomplete knowledge of functional networks, e.g., kinase–substrate annotations. To this end, construction of more comprehensive and detailed networks can potentially enhance the utility of missing sites in improving kinase activity inference. Overall, we recognize this as an important direction for future research.

## Methods

**Problem definition**. Kinase activity inference can be defined as the problem of predicting changes in kinase activity based on observed changes in the phosphorylation levels of substrates. Formally, let $K = \{k_1, k_2, ..., k_m\}$ denote a set of kinases and $S = \{s_1, s_2, ..., s_n\}$ denote a set of phosphorylation sites. For these kinases and phosphosites, a set of annotations are available, where $S_i \subseteq S$ denotes the set of substrates of kinase $k_i$, i.e., $s_j \in S_i$ if kinase $k_i$ phosphorylates site $s_j$.

In addition to the annotations, we are given a phosphorylation dataset representing a specific biological context. This dataset can be represented as a set of quantities $q_j$ for $1 \leq j \leq n$, where $q_j$ denotes the change in the phosphorylation level of phosphosite $s_j \in S$. Usually, $q_j$ represents the log-fold change of the phosphorylation level of the site between two sets of samples representing different conditions, phenotypes, or perturbations. The objective of kinase activity inference is to integrate the annotations and the specific phosphorylation data to identify the kinases with significant difference in their activity between these two sets of samples. In the below discussion, we denote the inferred change in the activity of kinase $k_i$ as $\hat{a}_i$. Since existing kinase activity inference methods are unsupervised, many activity inference methods also compute a $p$ value to assess the statistical significance of $\hat{a}_i$ for each kinase.

**Background**. *Mean (baseline)*: this is a simple method that represents the activity of a kinase as the mean phosphorylation (log-fold change) of its substrates:

$$\hat{a}_i^{(\text{mean})} = \frac{\sum_{s_j \in S_i} q_j}{|S_i|}. \tag{2}$$

where $|S_i|$ is the number of substrates of kinase $k_i$.

*Z-score*: this method normalizes the mean phosphorylation of the substrates to reflect statistical significance.

$$\hat{a}_i^{(z-\text{score})} = \frac{\sum_{s_j \in S_i} q_j}{\sigma \sqrt{|S_i|}} = \frac{\sqrt{|S_i|}}{\sigma} \hat{a}_i^{(\text{mean})}, \tag{3}$$

where $\sigma$ is the standard deviation of phosphorylation (log-fold changes) across all phosphosites.

*Linear model*: in this model, the phosphorylation of a site is modeled as summation of the activities of kinases that phosphorylate the site:

$$q_j = \sum_{\substack{\text{for all kinases } i \\ \text{phosphorylating site } j}} a_i \tag{4}$$

where $a_i$ is variable representing the activity of kinase $k_i$. To infer the kinase activities, least squares optimization function with ridge regularization is used:

$$\hat{a}^{(\text{linear})} = \underset{a}{\text{argmin}} \left\{ \sum_{s_j \in S} \left( q_j - \sum_{k_i \in K_j} a_i \right)^2 + \lambda ||a||^2 \right\}, \tag{5}$$

where $K_j$ denotes the set of kinases that phosphorylate site $s_j$, and $\lambda$ is an adjustable regularization coefficient. The first term in the objective function (squared loss) ensures that the inferred kinase activities are consistent with the phosphorylation levels of their substrates, whereas the second term (regularization) aims to minimize the overall magnitude of inferred kinase activities. In all experiments, we utilize a regularization coefficient of $\lambda = 0.1$ as previously done in ref. [39].

*GSEA*: to infer the activity of a kinase, this method assesses whether the substrates of the kinase are more enriched compared to other phosphosites in terms of their phosphorylation. To compute the enrichment score, the sites are first ranked based on their absolute fold changes. For each kinase $k_i$, a running sum is computed based on the ranked list of sites. The running sum increases for each site $s_j \in S_i$ (i.e., $s_j$ is a known substrate of $k_i$), and decreases for each site $s_j \notin S_i$ (i.e., $s_j$ is not a known substrate of $k_i$). The maximum deviation of this running sum from zero is used as the enrichment score of a kinase. The statistical significance of this enrichment score is assessed using a permutation test. Namely, fold changes of sites are permuted 10,000 times and enrichment scores are computed for each. The $p$ value for a kinase is then computed as the number of permutations with higher enrichment score than observed. As the predicted activity of a kinase, $-\log10$ of this $p$ value is used.

**Phospho-proteomics data preprocessing**. Following the footsteps of previous studies[18,39], we apply some quality control steps to the phospho-proteomics data that is used for benchmarking: (i) we restrict the analysis to mono-phosphorylated peptides that are mapped to canonical transcripts of Ensembl, (ii) we average the

log-fold changes of technical replicates, as well as peptides that are mapped to the same Ensembl position (even if the exact peptides sequences are not identical), and (iii) we filter out the peptides that are identified in only a single study to reduce the amount of false–positive phosphosites, (iv) we restrict the analysis to perturbations in the gold standard with >1000 phosphosite identifications (which leaves 81 perturbations). Finally, we exclude a hybrid perturbation (i.e., a mixture of both an activator and an inhibitor) from our analysis. As a result of these steps, we obtain 53,636 sites identified in at least one of 80 perturbations. For these 80 perturbations, there are 128 kinase-perturbation annotations for 25 different kinases.

**Computing benchmarking metric (top-*k*-hit).** To compute the $P_{hit}(k)$ metric (read "top-*k*-hit"), we apply the following procedure:

1. For each perturbation separately, we rank the kinases based on their absolute activities predicted by the inference method.
2. For each perturbation, we consider the top $k$ kinases with highest predicted activity and compare them with the "true" perturbed kinases in gold standard.
3. If any of the top $k$ kinases is a true kinase (i.e., a kinase that is perturbed in the experiment), we consider the inference method to be successful (i.e., a hit) for that perturbation.
4. We compute the percentage of perturbations with successful predictions and report this quantity as $P_{hit}(k)$. Since the gold standard dataset is incomplete, $P_{hit}(k)$ metric serves as a minimum bound on the expected probability of discovering an upregulated or downregulated kinase if top $k$ kinases predicted by the inference method were to be experimentally validated.

**Robust kinase activity inference**

*Heterogeneous network model.* RoKAI uses a heterogeneous network model in which nodes represent kinases and/or phosphosites. The edges in this network represent different types of functional association between kinases, between phosphosites, and between kinases and phosphosites. Namely, RoKAI's functional network consists of the following types of edges:

- Kinase–substrate associations: an edge between a kinase $k_i$ and site $s_j$ indicates that $k_i$ phosphorylates $s_j$. These kinase–substrate associations obtained from PhosphositePlus[32], representing 3877 associations between 261 kinases and 2397 sites.
- Structure distance evidence: this type of edge between phosphosites $s_i$ and $s_j$ represents the similarity of $s_i$ and $s_j$ on the protein structure. We obtain structure distance evidence from PTMcode[33], which contains 7821 unweighted edges between 8842 distinct sites. Note that, in this network, a large portion of the edges (7037 edges) are intra-protein.
- Coevolution evidence: this type of edge between phosphosites $s_i$ and $s_j$ indicates that the protein sequences straddling $s_i$ and $s_j$ exhibit significant coevolution. We obtain this coevolution network from PTMcode which contains 178,029 unweighted edges between 19,122 distinct sites. After filtering the sites for rRCS ≥ 0.9 provided by PTMcode, 41,799 edges between 8342 distinct sites remain. Note that, 3516 of these edges overlap with the structural distance network. Thus, when coevolution and structural distance networks are used together, these 3516 overlapping edges are considered to have a weight of 2.
- PPI: an edge between kinases $k_i$ and kinase $k_j$ represents a PPI between $k_i$ and $k_j$. We use the PPI network obtained from STRING[34]. As the edge weights, we utilize the combined scores provided by STRING. Overall, the kinase–kinase interaction network contains 13,031 weighted edges (weights ranging from 0 to 1) between 255 distinct kinases.

*Network propagation.* Let $\mathcal{G}(\mathcal{V}, \mathcal{E})$ represent RoKAI's heterogeneous functional network, where $\mathcal{V} = K \cup S$ and $\mathcal{E}$ contains four types of edges as described above. To propagate phosphorylation levels of sites over $\mathcal{G}$, we utilize an electric circuit model (illustrated in Fig. 1). In this model, each node $n_i \in \mathcal{V}$ (kinase or phosphosite) has a node potential $v_i$. Each edge $e_{ij} \in \mathcal{E}$ (which can be a kinase–substrate association, kinase–kinase interaction or association between a pair of phosphosites) has a conductance $c_{ij}$ that allows some portion of the node potential $v_i$ of node $n_i$ to be transferred to node $n_j$ in the form of a current $I_{ij}$:

$$I_{ij} = \left(v_i - v_j\right)c_{ij} \qquad (6)$$

As seen in the equation, the current $I_{ij}$ carried by an edge is proportional to its condundance and the difference in node potentials. In our model, we use the weights available in the corresponding networks to assign conductance values to the edges.

We model the phosphorylation level of a site $s_j$ that is identified in the experiment as a current source $I_j = q_j$ connected to the reference node (representing the control sample) with a unit conductance. This ensures that the node potential $v_j$ of site $s_j$ is equal to its phosphorylation level $q_j$, if it is not connected to any other nodes. This is because the current incoming to a node is

always equal to its outgoing current:

$$\text{Incoming current} = \text{Outgoing current}$$

$$q_i = v_i + \sum_{(i,j) \in E} (v_i - v_j)c_{ij}, \quad \text{if } n_i \text{ has quantification} \qquad (7)$$

$$0 = \sum_{(i,j) \in E} (v_i - v_j)c_{ij}, \quad \text{if } n_i \text{ does not have quantification}$$

Observe that, in this model, the nodes without measured phosphorylation levels (sites that are not identified in an MS run or kinases) act as a bridge for connecting (and transferring phosphorylation levels between) other nodes. This is an important feature of RoKAI as it allows incorporation of unidentified phosphosites in the network model.

To compute the node potentials for all nodes in the network, we represent Eq. (7) as a linear system:

$$\mathbf{Cv} = \mathbf{b} \qquad (8)$$

$$C_{ij} = \begin{cases} 1 & \text{if } i = j \text{ and } n_i \text{ has quantification} \\ c_{ij} & \text{if } i \neq j \text{ and } n_i n_j \in \mathcal{E} \\ 0 & \text{otherwise} \end{cases} \qquad (9)$$

$$b_i = \begin{cases} q_i & \text{if } n_i \text{ has quantification} \\ 0 & \text{otherwise} \end{cases} \qquad (10)$$

Thus, the node potentials $\mathbf{v}$ can be computed using linear algebra as follows:

$$\mathbf{v} = (\mathbf{C}^\top \mathbf{C})^{-1} \mathbf{C}^\top \mathbf{b} \qquad (11)$$

Note that, to make the matrix inversion numerically stable, we add a small $\tau = 10^{-8}$ to the diagonals of the matrix $\mathbf{C}$.

Once node potentials are computed, we output the propagated phosphorylation levels for identified sites as:

$$\hat{q}_j = v_j. \qquad (12)$$

These propagated phosphorylation levels $\hat{q}_j$ are used as input to kinase activity inference algorithms to obtain the inferred activity of kinases.

**Reporting summary.** Further information on research design is available in the Nature Research Reporting Summary linked to this article.

## Data availability
We obtain the benchmarking data from publicly available datasets of previous studies[18,39] (http://phosfate.com/download.html). We obtain the kinase–substrate annotations from PhosphositePlus[32] (http://www.phosphosite.org/staticDownloads). We obtain the human protein–protein interaction network from STRING[34] (http://string-db.org/cgi/download.pl). We obtain the coevolution and structure distance evidence between phosphosites from PTMcode33 (http://ptmcode.embl.de/data.cgi). We obtain the predicted kinase–substrates edges by NetworKin[28] (https://networkin.info/download.shtml). The materials (code and data) to reproduce the results are available in figshare (https://doi.org/10.6084/m9.figshare.12644864).

## Code availability
The source code of RoKAI and the custom scripts used for the analyses in Matlab are available online (https://github.com/serhan-yilmaz/RoKAI, http://compbio.case.edu/omics/software/rokai).

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

## Acknowledgements
This work was supported in part by the National Library Of Medicine of the National Institutes of Health under award number R01-LM012980. The content is solely the responsibility of the authors and does not necessarily represent the official views of the National Institutes of Health.

## Author contributions
S.Y. identified, designed and implemented algorithms, developed software, performed testing and validation, generated results and visualization, and drafted the manuscript. M.A. helped with the identification of data sources, and formulation of the kinase activity inference problem; also provided critical feedback on the manuscript. D.S. provided input on the technical issues related to mass spectrometry-based proteomics. A.E.C. contributed to the conception of the idea and provided critical feedback on the development of algorithms and preparation of the manuscript. M.R.C. provided insights on phospho-proteomics, kinase activity inference, and validation of predictions, contributed to the funding of the project, and provided critical feedback on the manuscript. M.K. conceived the general idea for the problem and algorithms, supervised algorithm development and testing, secured funding for the project, and edited the manuscript.

## Competing interests
The authors declare no competing interests.
