## [Peer Review File · Nature Communications]

Reviewers' Comments:

Reviewer #1:

Remarks to the Author:

This paper presents a new network-based approach to infer kinase activity from phosphoproteomic data. This problem is of interest to the community and is an active area of research. The method, RoKAI, employs an electric circuit model on a heterogeneous dataset to leverage complementary information for accurate kinase activity prediction. The authors illustrate multiple important aspects of the kinase activity inference problem - bias in annotated data exists that has served prior statistical inference methods, different types of information offers non-redundant and predictive power, and using kinases with missing information in the network helps with inference. Overall, this paper was a pleasure to read. I have a few comments that will help the context and interpretation of this work.

1. RoKAI uses an electric circuit model. Others have adopted a similar model for molecular networks, albeit working on different problems. Some that come to mind are eQED and a method to infer causal genes and dysregulated pathways. Incorporating citations such as these would help put the model's contribution in context.

Suthram et al. eQED: an efficient method for interpreting eQTL associations using protein networks. *Mol. Sys. Biol.* 2008.

Kim et al. Identifying Causal Genes and Dysregulated Pathways in Complex Diseases. *PLOS Comp. Biol.* 2011.

2. The paper uses the 25 annotated kinases that have least one perturbation. It was challenging for me to find where this number came from (the last sentence of Section 4.3). Also, the 80 perturbations in Section 4.3 seem inconsistent with the 91 perturbations in Section 2.3. Since these 25 annotated kinases are crucial for much of the results, listing them (e.g. in a supplementary table) and clearly explaining where they come from early in the document would be useful.

3. The paper does a good job of investigating the contributions from the individual datasets in the face of missing data. The authors also show how RoKAI consistently improves upon existing methods in Figure 5 for different values of k for their top- k -hit measure. The lines showing the average number of substrates in the top- k predictions for Z-Score may be misleading. At 50% missingness, what is the distribution of substrates for all kinases? I wonder if $k=2$ is just reporting the highest-degree kinases (which is implied in Fig 1), and the decrease is simply because there are no other kinases of high degree. A third line that shows the upper bound of the average number of substrates would help determine if this negative correlation is a limitation of the data (e.g., take the k kinases with the most number of substrates).

4. RoKAI models missing phosphosite quantification, which seems to be a big difference from previous methods. Figure 6 shows exactly the experiment I was looking for to see how this affects prediction. While there is a difference between Type I and Type II networks, I am surprised by how small this difference is. The difference at 50% missingness doesn't seem statistically significant. This may be due to the fact that, at 50% missingness, you are removing the links that have the potential to bridge this information. The violin plots at 0% missingness would be useful in confirming this intuition.

5. The Discussion is noticeably light, and there are many things to highlight from this work. In addition to the points in this review, the authors claim that despite this being an active area, KSEA is still widely used. How will RoKAI be useful when other state-of-the-art methods have not been widely adopted? The website interface and open-source data could be discussed here (though Matlab is proprietary, it is widely used).

Minor:

- The paper models phosphosites as independent; how biologically realistic is this? Could RoKAI

account for dependency among phosphosites on the same substrate?

- The authors use the STRING "combined" channel, when not all channels may be relevant to kinase activity inference. How does the method change when using only the "experimental" channel?
- The "top-k-hit" measure seems quite generous, and much of the analyses are presented with top-k-10. While Figure 5 illustrates the effect of k, the measure (compared to other approaches) deserves further discussion.
- Figure 7 is difficult to parse: the different shades of blue nodes & the thin vs. thick edges are hard to read. The middle two columns are very small. Finally, the visualization in the right column is odd since "Overall Effect" is simply a function of the other three axes.
- There are repetitive sections in the Results and the Methods.

Reviewer #2:

Remarks to the Author:

The authors present a new method for inference of Kinase Activity that incorporates the functional neighbourhood in the calculation and uses a new network propagation algorithm based on an electric circuit model. Overall they perform an excellent benchmarking of existing methods, highlighting also the effect of prior knowledge bias on the performance of different methods and show that their method is robust, performs well and is affected less than others by this bias. The paper was a pleasure to read and all analyses are done and explained very well. Both the method and the benchmarking approach will be welcomed in the community, no doubt. Especially the emphasis on evaluating, while removing the effect of prior knowledge bias will enable better future choices of methods for kinase activity inference. I have no major issues with this work, other than the fact that the website for the web server is not working at the moment and so I can't actually play around with the analysis/data that they present.

Optional suggestions for improvement could be to include sites that match known PSSMs that represent kinase-recognition sites, even if they are not known substrates. And an optional application to demonstrate the value of this work, could be to apply it on a large perturbation dataset such as the Hijazi et al, 2020, Nature Biotechnology and show that they can identify the perturbed kinase as well as known off target effects of these drugs. I guess this can be done already in the datasets they use in their benchmarking as well.

It would also be nice if the code was provided in a language that is free and available to all, rather than MATLAB, such as Python or R.

Very nice work overall.

Revision Responses

Manuscript Title: Robust Inference of Kinase Activity Using Functional Networks

Manuscript ID: NCOMMS-20-17440

Authors: Serhan Yılmaz, Marzieh Ayati, Daniela Schlatzer, A. Ercüment Çiçek, Mark R. Chance, Mehmet Koyutürk

We would like to thank the reviewers for their careful review of our work, their appreciation of our contribution, and their constructive feedback. Please see our response to each concern raised by the reviewers.

REVIEWER COMMENTS

Reviewer #1 (Remarks to the Author):

This paper presents a new network-based approach to infer kinase activity from phosphoproteomic data. This problem is of interest to the community and is an active area of research. The method, RoKAI, employs an electric circuit model on a heterogeneous dataset to leverage complementary information for accurate kinase activity prediction. The authors illustrate multiple important aspects of the kinase activity inference problem - bias in annotated data exists that has served prior statistical inference methods, different types of information offers non-redundant and predictive power, and using kinases with missing information in the network helps with inference. Overall, this paper was a pleasure to read. I have a few comments that will help the context and interpretation of this work.

Response: We thank the reviewer for their constructive and positive feedback. We greatly appreciate their comments, questions, and suggestions to make the paper stronger.

1. RoKAI uses an electric circuit model. Others have adopted a similar model for molecular networks, albeit working on different problems. Some that come to mind are eQED and a method to infer causal genes and dysregulated pathways. Incorporating citations such as these would help put the model's contribution in context:

- Suthram et al. eQED: an efficient method for interpreting eQTL associations using protein networks. Mol. Sys. Biol. 2008.
- Kim et al. Identifying Causal Genes and Dysregulated Pathways in Complex Diseases. PLOS Comp. Biol. 2011.

Response: We thank the reviewer for pointing out these papers. Besides the differences in the application, We cite these papers in the Introduction of the revised manuscript and discuss the similarities and differences as follows:

“In order to increase the coverage of network propagation, we develop an electric circuit based model (Doyle and Snell, 1984; Cowen et al., 2017) that is specifically designed to incorporate missing sites not identified by MS. While RoKAI does not impute phosphorylation levels for unidentified sites (i.e., it is not intended to fill in missing data), it uses these sites to bridge the functional connectivity among identified sites. Similar electric circuit based models have been employed in the analysis of expression quantitative trait loci (eQTL) to identify causal genes and dysregulated pathways (Suthram et al., 2008; Kim et al., 2011). However, one important distinction is that the electric circuit model in RoKAI does not aim to uncover intermediate nodes between select target nodes, rather, it propagates all available quantifications over the network in order to reduce the noise by capturing consistent changes in the functional neighborhood of every node.”

2. The paper uses the 25 annotated kinases that have least one perturbation. It was challenging for me to find where this number came from (the last sentence of Section 4.3). Also, the 80 perturbations in Section 4.3 seem inconsistent with the 91 perturbations in Section 2.3. Since these 25 annotated kinases are crucial for much of the results, listing them (e.g. in a supplementary table) and clearly explaining where they come from early in the document would be useful.

Response: Even though there were 91 perturbations with at least one kinase-perturbation annotation in the dataset, we restricted the analysis to perturbations with at least 1000 identified phosphosites (as done in the analyses of Ochoa et al. 2016). Additionally, we excluded a hybrid perturbation (i.e., a mixture of both an activator and an inhibitor) from our analysis thinking that the available kinase-perturbation annotations for that perturbation could be unreliable. These steps resulted in 80 perturbations. We now realize that these steps (as well as the 25 kinases with annotations) were not mentioned in the description of the dataset in the results section.

In the revised version, we included a supplementary data table that contains a list of these 25 annotated kinases and updated the “*Benchmarking Setup*” section in the results to clarify these points as follows:

“*Benchmarking data:* Ochoa et al. (2016) compiled phospho-proteomics data from a comprehensive range of perturbation studies and used these data to comprehensively benchmark the performance of kinase activity inference methods (Hernandez-Armenta et al., 2017). This benchmark data brings together 24 studies spanning 91 perturbations that are annotated with at least one up-regulated or down-regulated kinase. In each of the studies, the phosphorylation levels of phosphosites are quantified using mass spectrometry. After applying quality control steps (as described in the methods), we analyze a subset of this dataset encompassing 80 perturbations and 53636 phosphosites identified in at least one of these 80 perturbations. Overall, for these 80 perturbations, there are 128 kinase-perturbation annotations (which is considered gold standard) for 25 different kinases (listed in Supplementary Data 1). In our computational experiments, we use this dataset to assess the robustness of existing kinase activity inference methods and validate our algorithms.”

Additionally, we have updated the “*phospho-proteomics data preprocessing*” section in methods as follows:

“Following the footsteps of previous studies, we apply some quality control steps to the phospho-proteomics data that is used for benchmarking:

(i) we restrict the analysis to mono-phosphorylated peptides that are mapped to canonical transcripts of Ensembl, (ii) we average the log fold changes of technical replicates as well as peptides that are mapped to the same Ensembl position (even if the exact peptides sequences are not identical), and (iii) we filter out the peptides that are identified in only a single study to reduce the amount of false-positive phosphosites, (iv) we restrict the analysis to perturbations in the gold standard with more than 1000 phosphosite identifications (which leaves 81 perturbations). Finally, we exclude a hybrid perturbation (i.e., a mixture of both an activator and an inhibitor) from our analysis. As a result of these steps, we obtain 53636 sites identified in at least one of 80 perturbations. For these 80 perturbations, there are 128 kinase-perturbation annotations for 25 different kinases.”

3. The paper does a good job of investigating the contributions from the individual datasets in the face of missing data. The authors also show how RoKAI consistently improves upon existing methods in Figure 5 for different values of k for their top- k -hit measure. The lines showing the average number of substrates in the top- k predictions for Z-Score may be misleading. At 50% missingness, what is the distribution of substrates for all kinases? I wonder if $k=2$ is just reporting the highest-degree kinases (which is implied in Fig 1), and the decrease is simply because there are no other kinases of high degree. A third line that shows the upper bound of the average number of substrates would help determine if this negative correlation is a limitation of the data (e.g., take the k kinases with the most number of substrates).

Response: We have updated Figure 5 to include a black line that shows the upper bound of the average number of substrates. We agree with the reviewer that addition of this line provides context for the interpretation of the average of substrates in the top k predictions (specifically for the apparent negative trend in Z-Score and GSEA with respect to k). Indeed, we observe that even though the average of the kinases predicted by the methods do not reach the upper bound (even for low k values like 2), the average number of substrates is expected to go down as a function of k and the curve for each method follows a trajectory similar to that of the upper-bound.

4. RoKAI models missing phosphosite quantification, which seems to be a big difference from previous methods. Figure 6 shows exactly the experiment I was looking for to see how this affects prediction. While there is a difference between Type I and Type II networks, I am surprised by how small this difference is. The difference at 50% missingness doesn't seem statistically significant. This may be due to the fact that, at 50% missingness, you are removing the links that have the potential to bridge this information. The violin plots at 0% missingness would be useful in confirming this intuition.

Response: We were also surprised by the relatively small difference between Type I and Type II networks. To test whether this is due to the removal of kinase-substrate annotations, we investigated the behaviour at 0% missingness as the reviewer suggests (this result is shown in the 0% missingness point(s) for the robustness plot in Figure 6). However, we observed a small difference between Type I and Type II even at 0% missingness (Type I: 0.35, Type II: 0.363 top-10-hit probability on average), as compared to the improvement provided by RoKAI Type I over the baseline. We hypothesize that the reason behind the modesty of improvement might be biological redundancy i.e., sites that are reached by missing, bridge nodes may already be covered by regular paths consisting of identified nodes. We have updated the discussion to reflect these points (please see below).

In addition, we updated figure 6 and included a curve/violin plot for the “without RoKAI” performance in order to provide a reference level for the differences.

Paragraph added to Discussion:

“A key motivation in developing RoKAI was to utilize the missing sites without quantifications by keeping them as bridges in the network (thus, increasing the overall coverage of the network). In our experiments, we indeed observe a consistent improvement for incorporating missing sites (as compared to disregarding them completely). However, contrary to our expectation, the magnitude of this improvement is rather modest. We hypothesize that this may be because of (i)

biological redundancy i.e., sites that are reached by missing, bridge nodes may already be covered by other paths consisting of identified nodes, (ii) our incomplete knowledge of functional networks e.g., kinase-substrate annotations. To this end, construction of more comprehensive and detailed networks can potentially enhance the utility of missing sites in improving kinase activity inference. Overall, we recognize this as an important direction for future research.”

5. The Discussion is noticeably light, and there are many things to highlight from this work. In addition to the points in this review, the authors claim that despite this being an active area, KSEA is still widely used. How will RoKAI be useful when other state-of-the-art methods have not been widely adopted? The website interface and open-source data could be discussed here (though Matlab is proprietary, it is widely used).

Response: As indicated in our responses to the reviewer's comments above, some of the points raised by the reviewer helped us enhance the Discussion. In addition to the points that were added to Discussion to address the reviewer's above comments, we updated the first two paragraphs of the discussion to discuss the potential utility of RoKAI and the benchmarking system:

“By comprehensively utilizing available data on the functional relationships among kinases, phospho-proteins, and phosphorylation sites, RoKAI improves the robustness of kinase activity inference to the missing annotations and quantifications. Its implementation is available as open-source in Matlab as well as a web tool (<http://rokai.io>) for easy accessibility. We expect that this will facilitate the identification of understudied kinases with only a few annotations and lead to the development of novel kinase inhibitors for targeted therapy of many diseases such as cancer, Alzheimer's disease, and Parkinson's disease. As additional functional information on cellular signaling becomes available, the inclusion of these information in functional networks utilized by RoKAI will likely further enhance the accuracy and robustness of kinase activity inference.

The introduced benchmarking setup provides the opportunity to explore and compare the predictions of a variety of inference algorithms in terms of their robustness to missing annotations. It also allows the estimation of how utilization of different functional networks would influence the inference process. These features can help enable researchers to understand the trade-offs between different kinase activity inference algorithms in terms of their robustness, accuracy, and biases. As a potential resource, we provide the materials (code and data) to reproduce our analysis results in figshare (doi:10.6084/m9.figshare.12644864) that the users can adapt to test different inference methods and/or networks. Using such a framework, we believe the users can make more informed decisions for follow-up studies.”

Minor:

- The paper models phosphosites as independent; how biologically realistic is this? Could RoKAI account for dependency among phosphosites on the same substrate?

Response: This is an excellent point. Although we did not directly model the sites on the same substrate within the input networks of RoKAI, in our view, there are two other ways to incorporate the dependency between sites on the same substrate outside of RoKAI:

1. *Through inference method:* While most of the existing methods for kinase activity inference model the phosphosites as independent, the linear model (implemented by Mischnik, et al. 2015 and Hernandez-Armenta et al. 2017) also aims to model the dependencies between sites in the inference. To be more specific, the dependencies between sites are used to inform the inference of the activities of the kinases with overlapping targets (though not necessarily the sites on the same substrate). Since RoKAI can be used in conjunction with any substrate based inference method, this can provide a partial way to model the dependencies in the same substrate.
2. *Through data normalization/preprocessing:* As an alternative way of modeling dependencies on the same substrate, some recent studies use protein expression of the substrate to normalize phosphorylation levels of the sites (Zhang, et al. 2016). Although this provides a direct way to take into account the dependencies between sites on the same substrate, the reported results on the effectiveness of this approach are not conclusive and it is overall unclear whether accounting for protein expression improves kinase activity inference (Arshad, et al. 2019).

Overall, we recognize that this is an important problem that needs significant further research and we acknowledge this issue in the Discussion of the revised manuscript as follows:

“An important consideration in kinase activity inference is the dependencies between phosphorylation levels of sites. Some inference methods take into account the dependency between sites that are targeted by the same kinase (Mischnik, et al. 2015 and Hernandez-Armenta et al. 2017). On the other hand, recent studies utilize protein expression to take into account the dependency between sites on the same protein by normalizing phosphorylation levels of the sites, but results on the effectiveness of this approach are not conclusive (Zhang, et al. 2016, Arshad, et al. 2019). Whereas, RoKAI implicitly considers the dependencies between sites using a functional network model. We recognize the explicit modeling of the dependencies as an important open problem that can further enhance the performance and reliability of kinase activity inference.”

Added References:

- Zhang, Hui, et al. "Integrated proteogenomic characterization of human high-grade serous ovarian cancer." *Cell* 166.3 (2016): 755-765.
- Arshad, Osama A., et al. "An integrative analysis of tumor proteomic and phosphoproteomic profiles to examine the relationships between kinase activity and phosphorylation." *Molecular & Cellular Proteomics* 18.8 suppl 1 (2019): S26-S36.

- The authors use the STRING "combined" channel, when not all channels may be relevant to kinase activity inference. How does the method change when using only the "experimental" channel?

Response: Per the reviewer's suggestions, we tested the STRING "experimental" channel and compared with the "combined" channel. First, we repeated our analysis to estimate the effect size (as in figure 3). When we replaced the combined channel with the experimental, we observed only a minor increase in the predictivity of the annotated kinases (from 0.18 to 0.19, see below) while the number of kinase-kinase interactions decreased by 24% (from 13,031 to 9,912 edges). Next, we performed a robustness analysis at 50% missingness and investigated the effect of the STRING channel on the RoKAI predictions for two networks (while using the mean substrate phosphorylation as the inference method). As can be seen below, we did not observe a considerable change in the prediction accuracy neither for KS+PPI network nor the combined network (KS+PPI+SD+CoEv).

- The "top-k-hit" measure seems quite generous, and much of the analyses are presented with top-k-10. While Figure 5 illustrates the effect of k, the measure (compared to other approaches) deserves further discussion.

Response: In our view, the main purpose of kinase activity inference is to prioritize kinases for additional consideration and ideally for experimental validation. Yet, in practice, it is typically very costly to experimentally validate more than a few kinases and it is infeasible to manually inspect more than a couple dozen. To take this consideration into account, we use the measure, "top-k-hit", that focuses on the top-k kinase predictions for small values of k. We used primarily k=10 in our experiments because, in our opinion, it represents a reasonable number of kinases

to put to additional scrutiny. We also considered other approaches that are employed in the literature like area-under ROC curve (AUROC) and precision at recall 0.5. Such measures additionally take into account the performance for higher values of k (i.e., they consider a higher number of predictions, including kinases that are less significant/active). We find such measures problematic because, even though they include the performance for a high number of predicted kinases in their calculation, it would not be possible for a potential user to inspect or use those predictions (e.g., it would not make a difference for a user whether a kinase is ranked 100th or 200th, yet this could make a considerable difference for measures like AUROC).

We have updated the “*Benchmarking metric*” section in Results to explain our reasoning behind the top-k-hit measure as follows:

“*Benchmarking metric*: The main purpose of kinase activity inference is to prioritize kinases for additional consideration and ideally for experimental validation. However, in practice, it is typically very costly to experimentally validate more than a few kinases and it is infeasible to manually inspect more than a couple dozen. Whereas, benchmarking approaches that are employed in the literature like area-under receiver operating characteristics curve (AUROC) and precision at recall 0.5 consider high number of predictions (k), including kinases that are less significant/active. We find such measures problematic because, even though they include the performance for a high number of predicted kinases in their calculation, it would not be practical for a potential user to inspect or use that many predictions. To take this consideration into account, we use a metric, “top-k-hit”, that focuses on the top-k kinase predictions for small values of k. Since the gold standard dataset is incomplete, this metric essentially serves as a minimum bound on the expected probability of discovering an up-regulated or down-regulated kinase if top k kinases predicted by the inference method were to be experimentally validated. In our experiments, we use k=10 (unless otherwise specified) since it represents a reasonable number of kinases to be put to additional scrutiny before experimental validation.”

- Figure 7 is difficult to parse: the different shades of blue nodes & the thin vs. thick edges are hard to read. The middle two columns are very small. Finally, the visualization in the right column is odd since “Overall Effect” is simply a function of the other three axes.

Response: Based on the reviewer’s suggestions, we made the following changes on Figure 3 (the reviewer’s reference to the figure as Figure 7 seems to be a typo) to make it clearer and easier to read:

- Removed the Q-Q plots in the middle and made the other plots larger.
- Removed the “overall effect” from the radar chart and added it as a bar-plot instead.
- Updated the colors/line widths to make different shades of edges more distinguishable.
- Updated the figure layout to accommodate the changes.

- There are repetitive sections in the Results and the Methods.

Response: We went over the methods and removed the parts that overlapped with the results section (particularly in description of existing inference methods). While doing that, we noticed and fixed a minor error in the equation of z-score (Eq. 3) where the normalization with number of substrates should have been within square root.

Reviewer #2 (Remarks to the Author):

The authors present a new method for inference of Kinase Activity that incorporates the functional neighbourhood in the calculation and uses a new network propagation algorithm based on an electric circuit model. Overall they perform an excellent benchmarking of existing methods, highlighting also the effect of prior knowledge bias on the performance of different methods and show that their method is robust, performs well and is affected less than others by this bias. The paper was a pleasure to read and all analyses are done and explained very well. Both the method and the benchmarking approach will be welcomed in the community, no doubt. Especially the emphasis on evaluating, while removing the effect of prior knowledge bias will enable better future choices of methods for kinase activity inference.

Response: We thank the reviewer for their review and enthusiasm for our paper.

I have no major issues with this work, other than the fact that the website for the web server is not working at the moment and so I can't actually play around with the analysis/data that they present.

Response: RoKAI web server is currently hosted on a PC and we believe that it was offline due to some connectivity issues when the reviewer tried to access the web server. Unfortunately, our efforts to migrate RoKAI to a dedicated server have been delayed by difficulties associated with the alignment of the security protocols of Matlab web applications and security requirements of our institution. At this point, we are monitoring the accessibility of the web service on a daily basis and the reviewer should be able to access the current version without any problems. We are also planning to implement RoKAI and the web server in R (as we explain in our other response below). The URL we provide (<http://rokai.io>) is adaptable and will redirect to the new server once we complete its implementation in R and migration to a dedicated server.

Optional suggestions for improvement could be to include sites that match known PSSMs that represent kinase-recognition sites, even if they are not known substrates.

Response: This is an excellent idea. Since many existing algorithms use PSSMs to predict kinase-substrate interactions, we decided to implement this idea by utilizing predicted kinase-substrate interactions in kinase activity inference. For this purpose, we used NetworkKIN, which makes its predictions available for download. NetworkKIN lists its predictions as motif-based (NetPhorest), interaction-based (STRING), or combined (using both motif and interaction information). Using these predictions, we considered two strategies for including predicted interactions in RoKAI's framework: (i) Use predicted interactions in both RoKAI's functional network and during kinase activity inference (this strategy is annotated as RoKAI+), and (ii) Use only the known kinase-substrate interactions (PhosphositePlus) in RoKAI's functional network but also include predicted interactions during kinase activity inference (this strategy is annotated as RoKAI). The results of this analysis are shown in the figure below. We added this figure to the revised manuscript as Figure 7.

We also added the following text into the Result section to present and discuss these results:

“For this analysis, we use the baseline method (mean phosphorylation) for the inference. To make the results comparable with our previous analysis (using only the known substrates in PhosphositePlus), we limit the analysis to the kinases with at least one known substrate identified in the perturbation experiments (this way, we keep the kinase set same as before). The results of this analysis are shown in Figure 7, Here, the x axis shows the number of predicted interactions included in the inference (i.e., as we go right on the x axis we apply a more relaxed threshold on the prediction score). Thus, the leftmost point (0 at x-axis) corresponds to the case where only confirmed interactions (PhosphositePlus) are used.

As expected, the inclusion of predicted interactions in kinase activity inference improves the performance for the baseline algorithm and the performance of RoKAI-enhanced inference stays above the baseline. However, we observe that the use of predicted interactions together with RoKAI does not improve the inference further (while there is some increase in performance with the inclusion of high-confidence predictions, the inclusion of lower-confidence predictions degrades the performance). In addition, the inclusion of predicted interactions within the RoKAI's functional network always results in less accurate inference. Taken together, these observations suggest that, since RoKAI already includes functional and structural information to compute propagated phosphorylation levels, the inclusion of predicted interactions that use similar information does not further enhance the accuracy of the inference.”

And an optional application to demonstrate the value of this work, could be to apply it on a large perturbation dataset such as the Hijazi et al, 2020, Nature Biotechnology and show that they can identify the perturbed kinase as well as known off target effects of these drugs. I guess this can be done already in the datasets they use in their benchmarking as well.

Response: This is an excellent suggestion, we thank the reviewer for identifying this important potential application and pointing out this important work.

We looked carefully into the dataset provided by Hijazi et al. (2020) and prior work by Klaeger et al. (Science, 2017) reporting experimentally identified off-target effects. Between the two datasets, we have identified 9 perturbations that overlap with the original perturbation dataset we used in our experiments (4 perturbation types, with multiple replicates): namely PLK1 Inhibitor BI2536, Quizartinib, PI3Ki GDC-0941, and (6 replicates for) Vemurafenib. For these

perturbations, we investigated the rankings of the known off-target kinases and compared the rankings of these off-target kinases based on kinase activity inference with and without RoKAI (note that, lower rankings indicate better results). These results are provided in the below table (ranking without RoKAI shown red, ranking with RoKAI shown green and in parenthesis).

Perturbation (Replicate)	Kinase: Without RoKAI Ranking (With RoKAI Ranking)
PLK1 Inhibitor BI2536	PLK3: 44 (30)
Quizartinib	PDGFRB: 138 (119), PDGFRA: 139 (120), RET: 17 (31)
PI3Ki GDC-0941	mTOR: 48 (29)
Vemurafenib (#1)	LYN: 140 (138), WNK3: 17 (154)
Vemurafenib (#2)	LYN: 73 (23), WNK3: 43 (121)
Vemurafenib (#3)	LYN: 115 (150), WNK3: 14 (144)
Vemurafenib (#4)	LYN: 39 (10), WNK3: 42 (107)
Vemurafenib (#5)	LYN: 45 (13)
Vemurafenib (#6)	LYN: 15 (3)
Vemurafenib (average)	LYN: 71 (56), WNK3: 29 (131)

As seen on the table, RoKAI provides improvement in the ranking of LYN (Vemurafenib), mTOR (GDC-0941), PLK3 (BI2536), PDGFRB, and PDGFRA (Quizartinib), but lowers the ranking of WNK3 (Vemurafenib) and RET (Quizartinib). Since this is a premature analysis that provides only anecdotal information on the performance of RoKAI in identifying off-target kinases due to low sample size, we decided to leave these results out of the manuscript. For future work, we will adopt RoKAI to the problem of identifying off-target kinases and comprehensively evaluate its performance on these datasets. To acknowledge this point, we revised the Discussion as follows:

“A noteworthy complication in perturbation studies that concern kinase activity inference is the effect of off-target kinases. While recent studies systematically identify off-target kinases in perturbation studies (Klaeger et al. 2017; Hijazi et al. 2020), the extension of kinase activity inference algorithms and tools like RoKAI to distinguish off-target effects remains an open problem that can advance many important applications like drug development.”

It would also be nice if the code was provided in a language that is free and available to all, rather than MATLAB, such as Python or R.

Response: This is an excellent suggestion, we agree with the reviewer. We have started working on implementing the code and the web service in R's shiny environment. Since this needs to be implemented from scratch and represents a well-defined implementation project, we decided to assign this project to an undergraduate student in our group. Thus, this will take a few months to be completed and tested. We ensured that the links in the revised manuscript are adaptable so that we can redirect to the updated website and codebase when the R version is available.

Very nice work overall

Response: We greatly appreciate the reviewers' positive evaluation and encouraging comments. Their constructive feedback has also helped greatly

Reviewers' Comments:

Reviewer #1:

Remarks to the Author:

The authors did a great job following up on the comments from my previous review, and I appreciate the additional experiments and discussion points that have been added to the paper. I have no additional comments.

Reviewer #2:

Remarks to the Author:

The authors have addressed all my comments including the optional ones. As long as they ensure that the website stays live I have no further concerns/comments.